# Optimal scaling laws in learning hierarchical multi-index models

**Leonardo Defilippis**[1]**, Florent Krzakala**[2]**, Bruno Loureiro**[1]**, and Antoine Maillard**[1,3]

[1]Departement d'Informatique, École Normale Supérieure, PSL & CNRS
[2]Information, Learning and Physics Laboratory, École Polytechnique Fédérale de Lausanne (EPFL)
[3]INRIA Paris

## Abstract

In this work, we provide a sharp theory of scaling laws for two-layer neural networks trained on a class of *hierarchical multi-index* targets, in a genuinely representation-limited regime. We derive exact information-theoretic scaling laws for subspace recovery and prediction error, revealing how the hierarchical features of the target are sequentially learned through a cascade of phase transitions. We further show that these optimal rates are achieved by a simple, target-agnostic spectral estimator, which can be interpreted as the small learning-rate limit of gradient descent on the first-layer weights. Once an adapted representation is identified, the readout can be learned statistically optimally, using an efficient procedure. As a consequence, we provide a unified and rigorous explanation of scaling laws, plateau phenomena, and spectral structure in shallow neural networks trained on such hierarchical targets.

## 1 Introduction

Despite the staggering practical success of neural networks, we still lack a predictive theory answering a deceptively simple question: given a structured learning problem, *how does the network adapts to the task, in what order are the relevant features in the data learned, and how these translate in statistical efficiency?* This question sits at the intersection of three active lines of research. First, the empirical observation of *neural scaling laws* (Kaplan et al., 2020; Brown et al., 2020; Hoffmann et al., 2022) suggest that the performance of large models scale as a power-law in the training resources, yet—with few exceptions—our mathematical understanding of this relationship remains largely confined to linear models or networks in the lazy regime. Second, recent literature of the training dynamics of neural networks increasingly suggest that feature learning is not a smooth process, but are associated to long *plateaus and abrupt transitions* in the risk, with features (or *concepts* in this context) appearing sequentially rather than all at once Saxe et al. (2014); Wei et al. (2022); Schaeffer et al. (2023). Third, empirical analyses of trained networks have uncovered robust regularities in how the learned representations manifest in the trained network weights, such as in their spectral structure, but without a first-principles explanation of *why* particular features are preferred or *when* they should emerge (Martin & Mahoney, 2021; Wang et al., 2023; Thamm et al., 2024).

This paper provides an end-to-end answer to these questions in a mathematically tractable—yet genuinely feature learning—setting of a two-layer neural network trained on *hierarchical multi-index* data, a class of structured supervised learning tasks where the target function depends on a hierarchical combination of functions of the covariates Ren et al. (2025); Ben Arous et al. (2025); Defilippis et al. (2025b). Here, hierarchical denote the fact that the direction are *ordered*, in the sense that their relative importance decay as a power-law with their index number. This leads to a quasi-sparsity of the target representation, with ordered hierarchy of feature strengths as is classical in signal processing Mallat (1999); Donoho (2006). In this task, learning is fundamentally *representation-limited*: the labels $y = g(\boldsymbol{W}_\star \boldsymbol{x})$ depend on a low-dimensional but unknown subspace $\mathrm{span}(\boldsymbol{W}_\star) \subset \mathbb{R}^d$ of the input space $\boldsymbol{x} \in \mathbb{R}^d$, and generalization hinges on discovering this subspace as well as possible from the data. Our goal is to turn this qualitative picture into sharp, quantitative predictions. More precisely, our **main contributions** are:

(i) **Optimal Bayes rates for feature recovery.** We derive the exact information-theoretic limits for recovering the features subspace $\mathrm{span}(\boldsymbol{W}_\star)$ from $n = \Theta(d)$ samples. Further, we precisely characterize the associated optimal mean-squared error rates (a.k.a. *scaling laws*), unveiling the presence of cross-overs and plateaus regions that translate a fundamental trade-off between data-scarce and model-limited regions, depending on the hardness of the underlying hierarchical structure. Interestingly, these rates coincide with the minimax bounds for quasi-sparse recovery achieved by the LASSO Raskutti et al. (2011), and with previously conjectured results for shallow networks with quadratic activation Defilippis et al. (2024). Our analysis shows that these scalings are in fact universal for a broad class of hierarchical multi-index targets, well beyond the quadratic setting.

(ii) **A matching spectral algorithm and sequential feature emergence.** We introduce a simple, *target-agnostic* spectral estimator that provably achieves the Bayes-optimal rates derived above. The recovery proceeds sequentially: the $i$-th direction in the hierarchy becomes detectable at a sample complexity $n_i = \Theta(i^{2\gamma}d)$ —where $\gamma$ is the exponent controlling the hierarchy— leading to a cascade of sharp phase transitions.

(iii) **Learnability of the target function by neural networks.** Once the features/concepts subspace is identified, we show that learning the second-layer (readout) weights incurs no additional statistical bottleneck: the resulting prediction error matches the Bayes-optimal rate for subspace recovery.

Together, these results show that representation learning proceeds through a sequence of sharp phase transitions as the number of samples increases. New directions in the signal subspace emerge one after another, leading to plateaus and abrupt drops in the prediction error. This sequential emergence (to borrow the terminology of Wei et al. (2022); Schaeffer et al. (2023)) of features provides a precise theoretical underpinning for the empirically observed phenomenon of progressive concept learning in neural networks. Strikingly, the resulting phenomenology closely mirrors that recently uncovered for diagonal and quadratic neural networks using heuristic tools from statistical physics, where progressive feature emergence and distinct scaling regimes were observed (Defilippis et al., 2025b). Our contribution is to place this picture on firm theoretical ground and to substantially generalize it to a broader and more realistic class of models and targets.

## 2 SETTING

Consider a supervised regression problem with training data $\mathcal{D} = \{(\boldsymbol{x}_i, y_i) \in \mathbb{R}^{d+1} : i \in [n]\}$, which we assumed were drawn i.i.d. from a joint distribution over $\mathbb{R}^{d+1}$. Recall that the goal in regression is to learn a target function $f_\star(\boldsymbol{x}) = \mathbb{E}[y|\boldsymbol{x}]$ from the training data $\mathcal{D}$. In the following, we will be interested in particular structured tasks where the dependence of the target $f_\star$ on the covariates $\boldsymbol{x} \in \mathbb{R}^d$ is given by a hierarchical combination of low-dimensional subtasks. Mathematically, this is formalized by *hierarchical multi-index models*.

**Definition 2.1 (Hierarchical multi-index model)** *Let $\boldsymbol{W}_\star = (\boldsymbol{w}_k^\star \in \mathbb{R}^d)_{k \in [m_\star]}$ denote a family of $m_\star$ orthogonal vectors of norm $\|\boldsymbol{w}_k^\star\|^2 = d$. A hierarchical multi-index target is defined as*

$$f_\star(\boldsymbol{x}) = \sum_{k=1}^{m_\star} a_k^\star g_k(\langle \boldsymbol{w}_k^\star, \boldsymbol{x} \rangle), \tag{1}$$

*where $a_1^\star > a_2^\star > \ldots > a_{m_\star}^\star$ satisfy $\sum_{k=1}^{m_\star} (a_k^\star)^2 = 1$ and $g_k : \mathbb{R} \to \mathbb{R}$. Moreover, we say $f_\star$ is a scale-free hierarchical multi-index model if $a_k^\star = \Theta(k^{-\gamma})$ for some $\gamma > 0$.*

A few remarks are in order.

- As the name suggests, hierarchical multi-index models can be seen as a sum of $m_\star$ effectively one-dimensional tasks $g_k(z_k)$, with decreasing weight $a_k^\star$.
- More generally, multi-index models are functions of the type $f_\star(\boldsymbol{x}) = g(\boldsymbol{W}_\star\boldsymbol{x})$, where $g : \mathbb{R}^{m_\star} \to \mathbb{R}$ is known as the *link function* and

$$\boldsymbol{z} = \boldsymbol{W}_\star\boldsymbol{x} \in \mathbb{R}^{m_\star} \tag{2}$$

are known as the *indices*. They have been widely studied as statistical models for regression in the statistics literature Li (1991); Yuan (2011); Babichev & Bach (2018).
- More recently, multi-index models have gained in popularity in the machine learning theory literature as generative models for data, where they have been used to prove feature learning separation results for neural networks Damian et al. (2022); Abbe et al. (2023); Dandi et al. (2024).

- Scale-free hierarchical multi-index models were considered recently in Ren et al. (2025); Ben Arous et al. (2025); Defilippis et al. (2025b). Note that when $\gamma > 1/2$, the target is *quasi-sparse* in the index basis (Mallat, 1999; Donoho, 2006). This makes the model a suitable framework for investigating feature learning in neural networks, which are known to exhibit an implicit bias towards sparse estimators Gunasekar et al. (2017); Soudry et al. (2018); Andriushchenko et al. (2023).
- In scale-free hierarchical multi-index models with $\gamma < 1/2$, the coefficients must scale as $a_k^\star = \Theta(k^{-\gamma} m_\star^{\gamma-1/2})$ to guarantee the boundedness of the labels.

**Definition 2.2** *The training data* $\mathcal{D} = \{(\boldsymbol{x}_i, y_i) \in \mathbb{R}^{d+1} : i \in [n]\}$ *is drawn i.i.d. from a scale-free hierarchical multi-index model* $f_\star(\boldsymbol{x})$ *as in Def. 2.1, in particular*

$$y_i = f_\star(\boldsymbol{x}_i) + \sqrt{\Delta}\xi_i \tag{3}$$

*where* $\boldsymbol{x}_i \sim \mathcal{N}(0, 1/d\boldsymbol{I}_d)$, $\Delta > 0$ *is the noise variance and* $\xi_i \sim \mathcal{N}(0, 1)$.

Our results depend on mild assumptions on the link function, stated formally in Assumption 5.1.

In order to quantify the recovery of each individual index, or concept, we define following metric.

**Definition 2.3 (Matrix-MSE)** *Let* $\boldsymbol{w}_k^\star \in \mathbb{S}^{d-1}(\sqrt{d})$ *denote one of the target indices* $k \in [d]$. *We define the matrix-MSE associated to a predictor* $\boldsymbol{w} \in \mathbb{R}^d$ *as* $\mathrm{mse}_k(\boldsymbol{w}) := \frac{1}{d^2}\mathbb{E}\left[\|\boldsymbol{w}\boldsymbol{w}^\top - \boldsymbol{w}_k^\star\boldsymbol{w}_k^{\star\top}\|_F^2\right]$.

Additionally, we characterize the *weak recovery* transition for the index $k$, *i.e.* the minimum number of samples required by an estimator to correlate non-trivially with a specific concept.

**Definition 2.4 ($k-$critical threshold)** *Given an estimator* $\hat{\boldsymbol{w}}_k \in \mathbb{R}^d$ *of the signal direction* $\boldsymbol{w}_k^\star$ *(i.e. a measurable function of the training data* $\boldsymbol{X}, \boldsymbol{y}$*), the* $k-$*critical threshold is defined as the minimum sample complexity such that the estimator exhibits a finite overlap with* $\boldsymbol{w}_{\star,k}$, *namely* $\inf\left\{\alpha > 0 : d^{-1}|\langle\hat{\boldsymbol{w}}_k, \boldsymbol{w}_{\star,k}\rangle| = \Theta_d(1)\right\}$.

The recovery of $\mathrm{span}(\boldsymbol{W}^\star)$ is assessed by the sum of the squared errors for each individual direction, weighted by the contribution of each concept to the target variance.

**Definition 2.5 (Weighted MSE)** *Let* $\boldsymbol{W} \in \mathbb{R}^{m_\star \times d}$. *We define the weighted mean-squared error as*

$$\mathrm{MSE}_\gamma(\boldsymbol{W}) := \sum_{k \in K} (a_k^\star)^2 \mathrm{mse}_k(\boldsymbol{w}_k). \tag{4}$$

For the second goal — showing that neural networks can efficiently learn $f_\star$ — we will focus on two-layer neural networks:

$$f(\boldsymbol{x}; \boldsymbol{\Theta}) := \boldsymbol{a}^\top \sigma(\boldsymbol{W}\boldsymbol{x} + \boldsymbol{b}), \tag{5}$$

with weights $\boldsymbol{\Theta} := \left(\boldsymbol{a} \in \mathbb{R}^p, \boldsymbol{W} \in \mathbb{R}^{p \times d}\right)$ and a generic activation function $\sigma : \mathbb{R} \to \mathbb{R}$. As we shall see, two-layer neural networks can agnostically learn with respect to the model, i.e. without knowledge of the individual tasks $g_k$ nor the details in Def. 2.1. We quantify the generalization capacity of this model through its excess risk

$$R(\boldsymbol{\Theta}) = \mathbb{E}\left[(f_\star(\boldsymbol{x}) - f(\boldsymbol{x}; \boldsymbol{\Theta}))^2\right]. \tag{6}$$

# 3 MAIN RESULTS

## 3.1 BAYES-OPTIMAL RATES FOR FEATURE RECOVERY

As a first result, we characterize the information-theoretic limits for the recovery of $\mathrm{span}(\boldsymbol{W}_\star)$ in terms of the sample complexity $\alpha$ and the subspace dimension $m_\star$.

**Definition 3.1 (Optimal MSE)** *Let* $\mathcal{D} = (\boldsymbol{X}, \boldsymbol{y})$ *be drawn as in def. 2.2. Then, the optimal (weighted) mean-square error is achieved by the posterior average* $\mathrm{MMSE}_\gamma := \sum_{k=1}^{m_\star} \frac{(a_k^\star)^2}{d^2}\mathbb{E}\left[\|\mathbb{E}[\boldsymbol{w}_k\boldsymbol{w}_k^\top | \mathcal{D}] - \boldsymbol{w}_k^\star\boldsymbol{w}_k^{\star\top}\|_F^2\right]$.

By definition, $\mathrm{MMSE}_\gamma$ is a lower-bound for the smallest achievable weighted mean-squared error $\mathrm{MSE}_\gamma(\hat{\boldsymbol{W}})$ by any estimator $\hat{\boldsymbol{W}}$ that is a function of the dataset $\mathcal{D}$. Our first main result quantifies precisely the rates of $\mathrm{MMSE}_\gamma$ as $\alpha \gg 1$, which by construction define the optimal scaling laws for subspace reconstruction in the class of scale-free hierarchical multi-index models. A proof of this result is discussed in Appendix 5.

**Theorem 3.2 (Optimal scaling-laws)** *In the setting of Definitions 2.1, 2.2, under Assumption 5.1, for $\alpha, m_\star \gg 1$, the Bayes-optimal mean-squared error satisfies*

$$\mathrm{MMSE}_\gamma = \Theta_{\alpha, m_\star} \begin{cases} \min(\alpha^{-1+\frac{1}{2\gamma}}, \frac{m_\star}{\alpha}) & \gamma > 1/2, \\ \min(1, \frac{m_\star}{\alpha}), & \gamma < 1/2 \end{cases} \tag{7}$$

*Moreover, the $k$-critical threshold of the Bayes estimator satisfies $\alpha_k^{\mathrm{Bayes}} = \Theta_k(k^{2\gamma} m_\star^{\max((1-2\gamma),0)})$*

## 3.2 OPTIMAL AGNOSTIC SUBSPACE RECOVERY

While Theorem 3.2 provides a fundamental benchmark, an equally important question is whether the optimal reconstruction rates can be *efficiently* achieved by an algorithm which is agnostic of the underlying data distribution.

**Definition 3.3 (Spectral estimator)** *Given a pre-processing function $\mathcal{T} : \mathbb{R} \to \mathbb{R}$ on the labels, consider the symmetric random matrix*

$$\boldsymbol{T} = \sum_{i=1}^{n} \mathcal{T}(y_i) \boldsymbol{x}_i \boldsymbol{x}_i^\top, \tag{8}$$

*with spectrum $\lambda_1 \geq \lambda_2 \geq \ldots \geq \lambda_d$. Define $r := \min\{i \in \mathbb{N} \,|\, \lambda_{i+1} - \lambda_{i+2} < C/\sqrt{d}\}$, for some arbitrary constant $C$. We define the spectral estimator $\hat{\boldsymbol{W}}^{\mathrm{sp}} = (\hat{\boldsymbol{w}}_1, ..., \hat{\boldsymbol{w}}_r)^\top \in \mathbb{R}^{r \times d}$ where $\hat{\boldsymbol{w}}_k$ is the $k \in [r]$ eigenvector of $\boldsymbol{T}$ corresponding to $\lambda_k$, normalized such that $\|\hat{\boldsymbol{w}}_i\|^2 = d$.*

In Appendix 5.2.1 we discuss an interpretation of the spectral estimator in terms of a gradient descent algorithm. Crucially, we consider $\mathcal{T}$ to be data-agnostic, satisfying only the following condition.

**Assumption 3.4** *Given $Y \sim \mathcal{N}\left(\sum_{k=1}^{m_\star} a_k^\star g_k(z_k), \Delta\right)$ with $\boldsymbol{z} \sim \mathcal{N}(\boldsymbol{0}, \boldsymbol{I}_{m_\star})$, assume that $\mathbb{P}(\mathcal{T}(Y) = 0) < 1$ and $\mathcal{T}$ is bounded and there exists $\tau > 0$ such that $\tau = \inf\{c : \mathbb{P}(\mathcal{T}(Y) < c) = 1\}$.*

Our second main theoretical result shows that the data-agnostic spectral method defined above indeed achieves the optimal reconstruction rates of Theorem 3.2.

**Theorem 3.5** *In the setting of Definitions 2.1, 2.2, under Assumption 5.1, denote $\hat{\boldsymbol{W}} = (\hat{\boldsymbol{W}}^{\mathrm{sp}}, \boldsymbol{0}_{(m_\star - r) \times d}) \in \mathbb{R}^{m_\star \times d}$, where $\hat{\boldsymbol{W}}^{\mathrm{sp}} \in \mathbb{R}^{r \times d}$, $r \leq m_\star$, is the spectral estimator defined in 3.3, with $\mathcal{T}$ satisfying Assumption 3.4. Then, for $\alpha, m_\star \gg 1$, denoting the spectral estimator's $k$-critical threshold as $\alpha_k^{\mathrm{sp}}$, we have that $\mathrm{MSE}_\gamma(\hat{\boldsymbol{W}}) = \Theta_{\alpha, m_\star}(\mathrm{MMSE}_\gamma)$, and $\alpha_k^{\mathrm{sp}} = \Theta_k(\alpha_k^{\mathrm{Bayes}})$.*

Figure 2 shows the decay of the weighted MSE for three scale-free hierarchical model variants as a function of the sample complexity $\alpha$, based on finite-size experiments, using the spectral method introduced in Definition 3.3. Figure 3 illustrates the sequential emergence of new concepts as sample complexity grows, visualized as spikes detaching from the eigenvalue bulk of the matrix $\boldsymbol{T}$ in eq. 8.

## 3.3 LEARNING MULTI-INDEX WITH NEURAL NETWORKS

Finally, we turn our attention to the problem of learning hierarchical multi-index targets function with a two-layer neural network and show that the excess risk associated with a suitable training procedure described in Algorithm 1 in Appendix 6, achieves the optimal rates in Theorem 3.2.

**Theorem 3.6** *In the setting of Definitions 2.1, 2.2, under Assumption 5.1, with $g_k^\star$, $k \in [m_\star]$, polynomials of finite degree. There exist $n_0$ such that, for $n > n_0$, $p = \omega(n^{1/2})$, $\lambda = \Theta(n^{-1/2})$, the excess risk of eq. equation 6 for a two-layer neural network equation 5, with $\sigma$ bounded and continuous, trained according to Algorithm 1 satisfies*

$$R_{\mathrm{NN}} = \Theta(\mathrm{MMSE}_\gamma) + O(n^{-1/2}). \tag{9}$$

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

## 4   FURTHER RELATED WORKS —

Most theoretical work on neural scaling laws focus on effectively linear models, such as kernel methods and random features, where the generalization behavior can be characterized through spectral properties of fixed representations (Caponnetto & De Vito, 2007; Spigler et al., 2020; Cui et al., 2021; 2023; Defilippis et al., 2024; Atanasov et al., 2024; Bahri et al., 2024; Maloney et al., 2022; Paquette et al., 2024; Kunstner & Bach, 2025; Bordelon et al., 2020). Exceptions have focused either in the joint-training of both layers for linear networks Bordelon et al. (2024; 2025); Worschech & Rosenow (2024) or non-linear settings with fixed-features Wortsman & Loureiro (2025); Worschech & Rosenow (2024); Braun et al. (2025). A central difference is that these works introduce scaling through the covariate distribution, while in our work it is induced by the hierarchical nature of the task. Moreover, our work departs from this literature by addressing scaling laws in a genuinely non-linear, feature-learning regime.

Closer to us are a recent line of work analyzing two-layer networks with structured first-layer weights and hierarchical multi-index target (Ren et al., 2025; Ben Arous et al., 2025; Defilippis et al., 2025b). In particular, Ren et al. (2025); Ben Arous et al. (2025) studied one-pass SGD in this setting, with Ben Arous et al. (2025) focusing on the case of a quadratic neural network architecture. The main difference with this work is that we analyze full-batch ERM, characterizing the optimal scaling laws and showing that they can be achieved computationally by a gradient-descent like algorithm, thus providing a fundamental benchmark for the SGD rates in these works.

Complementary, Defilippis et al. (2025b) derived rates for ERM in quadratic neural networks trained on quadratic targets. Our results show that the rates from Defilippis et al. (2025b) are universal a large class of target functions (any generative exponent two functions in the language of Damian et al. (2024)), closing a gap between the purely quadratic setting and general hierarchical multi-index targets. The key underlying these universal scaling laws is the combination *quasi-sparsity* of the target representation, encoded by a heavy-tailed spectrum that induces an ordered hierarchy of feature strengths (Mallat, 1999; Donoho, 2006), and an implicit *rank-sparsity* bias, leading to LASSO-like behavior (Raskutti et al., 2011).

On a technical level, our work builds on the toolbox of approximate message passing (AMP) and its associated state evolution (Bayati & Montanari, 2011; Donoho et al., 2013; Javanmard & Montanari, 2013; Berthier et al., 2020; Zou & Yang, 2022; Feng et al., 2022; Gerbelot & Berthier, 2023; Dudeja et al., 2023; Erba et al., 2025). In particular, our results leverage the connection between AMP and Bayes-optimal estimation for single- Barbier et al. (2019) and multi-index Aubin et al. (2018); Troiani et al. (2025) models. Similarly, we build up on the literature on optimal spectral methods derived from AMP for single- (Mondelli & Montanari, 2019; Lu & Li, 2020; Maillard et al., 2022; Damian et al., 2024) and multi-index Kovačević et al. (2025); Defilippis et al. (2025a) functions.

## 5   PROOF SKETCHES OF THEOREMS 3.2 AND 3.5

In this section, we prove the information-theoretic results stated in Theorem 3.2. In particular, we derive matching bounds for the $\mathrm{MMSE}_\gamma$, defined 3.1, and the $k$-critical threshold $\alpha_k$ (Definition 2.4). Our results depend on the following additional assumption of the link functions.

**Assumption 5.1** *For each index $k \in [m_\star]$, $g_k$ is an even function. Moreover, for $z \sim \mathcal{N}(0, 1)$, for some constants $C, D > 0$, independent of $k, n, d, \alpha, m_\star$,*

$$D < \mathbb{E}_z[g_k^2(z)] < C, \quad D < |\mathbb{E}_z[g_k''(z)]| < C. \tag{10}$$

**Remark 5.2** *The parity assumption on $g_k$ ensures that learning $\mathrm{span}(\boldsymbol{W}_\star)$ is non-trivial, by ruling out linear correlations that would allow recovery at arbitrarily small sample complexity. Indeed, given $\boldsymbol{z}$ as in equation 2 and $y$ as in eq. 3, when $\mathbb{E}[\boldsymbol{z} \mid y] = 0$ a.s.—which holds here by parity—no efficient algorithm can even weakly recover $\mathrm{span}(\boldsymbol{W}_\star)$ below the critical threshold*

$$\alpha_c := \left( \sup_{\boldsymbol{M} \in \mathbb{S}_{m_\star}^+, \|\boldsymbol{M}\|_F = 1} \|\mathbb{E}_y \boldsymbol{G}(y) \boldsymbol{M} \boldsymbol{G}(y)\|_F \right)^{-1}, \tag{11}$$

*with $\boldsymbol{G}(y) := \mathbb{E}[\boldsymbol{z}\boldsymbol{z}^\top - \boldsymbol{I}_{m_\star} \mid Y = y]$ Barbier et al. (2019); Mondelli & Montanari (2019); Lu & Li (2020); Damian et al. (2024); Troiani et al. (2025).*

*The condition $\mathbb{E}_z[g_k'^2(z)] < C$ controls the label variance, while the lower bound on $\mathbb{E}_z[g_k''(z)]$ ensures detectability in the proportional regime $n = \Theta(d)$ (equivalently, a generative exponent equal to 2), so that the relative difficulty of each index is governed solely by the decay of $a_k^\star$.* [1]

*Hierarchical multi-index models with generative exponent equal to 2 includes the vast majority of cases of interest, while the class of models with exponent larger than 2 mostly includes fine-tuned examples Barbier et al. (2017); Damian et al. (2024).*

In Appendix 5.1, we analyze an oracle estimator that exploits additional information to achieve a weighted mean-squared error smaller than $\mathrm{MMSE}_\gamma$. Finally, in Appendix 5.2, we complete the proof by characterizing the scaling laws and $k$-critical thresholds of the spectral estimator defined in 3.3, which simultaneously establishes the results in Theorem 3.5.

## 5.1 Lower bound

**Definition 5.3 (Oracle Estimator)** *Consider a dataset $\mathcal{D} = \{(\boldsymbol{x}_i, y_i) \in \mathbb{R}^d \times \mathbb{R}\}_{i \in [n]}$, where the labels are generated by a multi-index model $y \sim \mathsf{P}(\cdot | \langle \boldsymbol{w}_1^\star, \boldsymbol{x} \rangle, \dots, \langle \boldsymbol{w}_{m_\star}^\star, \boldsymbol{x} \rangle)$, with weights $\boldsymbol{W}^\star$ drawn from a distribution $\mathsf{P}_{\boldsymbol{W}}$. We define the Oracle Estimator as the matrix $\boldsymbol{W}^{\mathrm{oracle}} \in \mathbb{R}^{m \times d}$ with rows*

$$\boldsymbol{w}_k^{\mathrm{oracle}} = \arg\min_{\boldsymbol{w}} \mathbb{E}\left[\|\boldsymbol{w}_k \boldsymbol{w}_k^\top - \boldsymbol{w}_k^\star \boldsymbol{w}_k^{\star\top}\|_F^2 \,|\, \mathcal{D}, \{\boldsymbol{w}_h\}_{h \neq k}\right], k \in [m_\star]. \tag{12}$$

Analogously to the Bayes-optimal estimator case, the weighed mean-squared error $\mathrm{MSE}_\gamma(\boldsymbol{W}^{\mathrm{oracle}})$ is lower bounded by

$$\mathrm{MMSE}_\gamma^{\mathrm{oracle}} = \frac{1}{d^2} \sum_{k=1}^{m_\star} (a_k^\star)^2 \mathbb{E}\left[\|\mathbb{E}[\boldsymbol{w}_k \boldsymbol{w}_k^\top | \mathcal{D}, \{\boldsymbol{w}_h^\star\}_{h \neq k}] - \boldsymbol{w}_k^\star \boldsymbol{w}_k^{\star\top}\|_F^2\right]. \tag{13}$$

As a consequence of the law of total variance

$$d^2 \, \mathrm{mmse}_k := \mathbb{E}\left[\mathrm{Cov}\left(\boldsymbol{w}_k \boldsymbol{w}_k^\top | \mathcal{D}\right)\right] \tag{14}$$

$$= \mathbb{E}\left[\mathrm{Cov}\left(\boldsymbol{w}_k \boldsymbol{w}_k^\top | \mathcal{D}, \{\boldsymbol{w}_h\}_{h \neq k}\right)\right] + \underbrace{\mathbb{E}\left[\mathrm{Cov}_{\{\boldsymbol{w}_h\}_{h \neq k}}\left(\mathbb{E}\left[\boldsymbol{w}_k \boldsymbol{w}_k^\top | \mathcal{D}, \{\boldsymbol{w}_h\}_{h \neq k}\right]\right)\right]}_{\geq 0} \tag{15}$$

$$\geq d^2 \, \mathbb{E}\left[\|\mathbb{E}[\boldsymbol{w}_k \boldsymbol{w}_k^\top | \mathcal{D}, \{\boldsymbol{w}_h^\star\}_{h \neq k}] - \boldsymbol{w}_k^\star \boldsymbol{w}_k^{\star\top}\|_F^2\right]. \tag{16}$$

Therefore,

$$\mathrm{MMSE}_\gamma^{\mathrm{oracle}} \leq \mathrm{MMSE}_\gamma. \tag{17}$$

We now focus on the specific setting of our interest defined in Section 2. For each index $k \in [m_\star]$, conditioning on $\{\boldsymbol{w}_h\}_{h \neq k}$, the problem of the optimal estimation of $\boldsymbol{w}_k^\star$ becomes statistically equivalent to the Bayes-optimal estimation given a dataset $\mathcal{D}_k := \{(\boldsymbol{x}_\nu, \overline{y}_\nu)\}_{\nu \in [n]}$, where the labels are generated by the single-index model

$$\overline{y}_i = a_k^\star g_k(\langle \boldsymbol{w}_k^\star, \boldsymbol{x}_i \rangle) + \sqrt{\Delta} \xi_i, \tag{18}$$

where $\xi_i$ is the same label noise as in the original dataset. We can now characterize the oracle estimator using the results from the literature on *single-index models*, in particular Barbier et al. (2019). In Appendix 7 we show that the information-theoretic weak recovery threshold for single-index models is a monotonic decreasing function of the signal-to-noise ratio. Combined with Assumption 5.1, this implies that the sequence of $k$-critical thresholds $\alpha_k$ is bounded by strictly increasing functions.[2] In particular, the first result in Corollary 7.4 implies that the sequence is bounded by

$$\alpha_k^{\mathrm{oracle}} = \Theta\left((a_k^\star)^{-2}\right) = \Theta(k^{2\gamma} m_\star^{(1-2\gamma)_+}), \tag{19}$$

---

[1] Our analysis can be easily adapted to the less restrictive assumption that there exists an integer $\beta \geq 1$ such that $D < |\mathbb{E}_Z[g_k^\beta(Z)(Z^2 - 1)]| < C$. This would result in a simple modification of the scaling laws and not affect the overall message of the present manuscript.

[2] If $g_k = g$, $\forall k$, the sequence $\alpha_k^{\mathrm{oracle}}$ itself is strictly increasing.

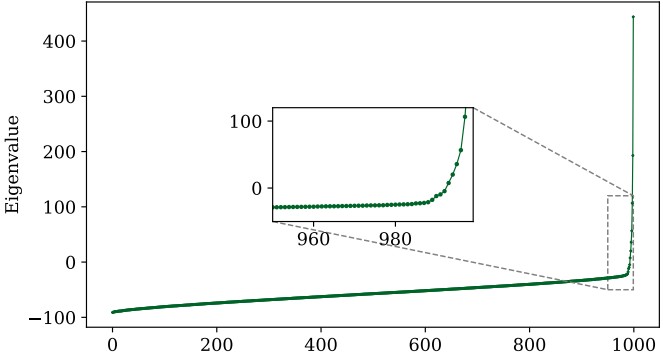

Figure 1: Empirical spectrum of $\boldsymbol{T}$ equation 8 with preprocessing $\mathcal{T}(y) = y/(1 + |y|)$, for a hierarchical multi-index model with $g_k(z) = \frac{1}{2}\mathrm{He}_2(z) + \frac{1}{2\cdot 4!}\mathrm{He}_4(z)$ and $\gamma = 1.3$. The covariates dimension is $d = 1000$, while the feature space dimension is $m_\star = 20$. The figure illustrates the change in scale of the eigenvalue gaps, transitioning from the informative spikes ($\Theta_d(1)$) to the uninformative bulk ($o_d(1)$). This behavior forms the basis of the selection method described in Def. 3.3.

with $(x)_+ = \max(0, x)$. Further, denoting by $k_\alpha$ the largest index $k$ such that $\alpha > \alpha_k$, i.e. $k_\alpha = \max(m_\star, \Theta(\alpha^{1/(2\gamma)}))$ for $\gamma > 1/2$ or $k_\alpha = \max(m_\star, \Theta(m_\star^{-1+1/(2\gamma)}\alpha^{1/(2\gamma)}))$ for $\gamma < 1/2$, and exploiting the second result in Corollary 7.4,

$$\mathrm{MMSE}_\gamma^{\mathrm{oracle}} = \left(\sum_{k=1}^{k_\alpha} + \sum_{k=k_\alpha+1}^{m_\star}\right)(a_k^\star)^2 \mathbb{E}\left[\|\mathbb{E}[\boldsymbol{w}_k\boldsymbol{w}_k^\top|\mathcal{D}, \{\boldsymbol{w}_h^\star\}_{h\neq k}] - \boldsymbol{w}_k^\star\boldsymbol{w}_k^{\star\top}\|_F^2\right] \quad (20)$$

$$\geq C\left(\sum_{k=1}^{k_\alpha}\frac{(a_k^\star)^2}{(a_k^\star)^2\alpha} + \sum_{k=k_\alpha+1}^{m_\star}(a_k^\star)^2\right) \quad (21)$$

$$= \begin{cases} \Theta\left(\alpha^{-1+1/(2\gamma)}\right), & \gamma > 1/2, \ \alpha \ll m_\star^{2\gamma} \\ \Theta\left(m_\star/\alpha\right), & \gamma > 1/2, \ \alpha \gg m_\star^{2\gamma} \\ \Theta\left(1\right), & \gamma < 1/2, \ \alpha \ll m_\star \\ \Theta\left(m_\star/\alpha\right), & \gamma < 1/2, \ \alpha \gg m_\star \end{cases} \quad (22)$$

Note that we have used

$$\sum_{k=k_\alpha+1}^{m_\star} k^{-2\gamma} = \begin{cases} \Theta(k_\alpha^{1-2\gamma}), & \gamma > 1/2, \\ \Theta(m_\star^{1-2\gamma}), & \gamma < 1/2. \end{cases} \quad (23)$$

In equation 21, the first term is a lower bound to the (weighted) mean-squared error of weakly recovered features, while the second corresponds to the underfitting contribution of the unlearned ones.

## 5.2 Upper Bound: Spectral Method

By definition, any estimator that is a function of the dataset only has a weighted mean-squared error larger than $\mathrm{MMSE}_\gamma$. In this section we consider the spectral method defined in 3.3. Note that, as the spectral method 3.3 retrieves $r \leq m_\star$ directions, we construct the full estimator by filling the remaining columns with zeros to evaluate $\mathrm{MSE}_\gamma$ in Def. 2.5. This zero-padding is harmless: the added zeros do not impact the derived scaling laws or the estimator's agnostic nature. The following Theorem, proven in Kovačević et al. (2025), is valid for generic Gaussian multi-index models with $m_\star$ indices. We refer to the original work for further details. In the rest of the section we denote by $\mathbb{E}$ the expected value with respect to $\boldsymbol{z} \sim \mathcal{N}(\boldsymbol{0}_{m_\star}, \boldsymbol{I}_{m_\star})$ and $y \sim \mathcal{N}\left(\sum_{k=1}^{m_\star}(a_k^\star)^2 g_k(z_k), \Delta\right)$.

**Theorem 5.4 (Theorem 4.1 in Kovačević et al. (2025))** *Let $\mathcal{T} : \mathbb{R} \to \mathbb{R}$ be a preprocessing function subject to Assumption 3.4 and $\boldsymbol{T}$ defined as in eq. 8. Let $t_1 \geq t_2 \geq \ldots \geq t_r \geq \tau$, for some $r \in [m]$, be all the solutions to*

$$\det\left(\alpha\mathbb{E}\left[\frac{(\boldsymbol{z}\boldsymbol{z}^\top - \boldsymbol{I}_m)\mathcal{T}(y)}{t - \mathcal{T}(y)}\right] - \boldsymbol{I}_p\right) = 0 \tag{24}$$

*such that*

$$t_k \geq \bar{t}_\alpha := \underset{t \geq \tau}{\arg\min}\ \zeta_\alpha(t), \quad \forall k \in [j], \tag{25}$$

*where*

$$\zeta_\alpha(t) := t\left(1 + \alpha\mathbb{E}\left[\frac{\mathcal{T}(y)}{t - \mathcal{T}(y)}\right]\right). \tag{26}$$

*Then, denote $\lambda_1^{\boldsymbol{T}}, \ldots, \lambda_{m_\star}^{\boldsymbol{T}}$ the largest $m_\star$ eigenvalues of $\boldsymbol{T}$. For the top $r$ eigenvalues it holds that*

$$\lambda_1^{\boldsymbol{T}}, \ldots, \lambda_r^{\boldsymbol{T}} \xrightarrow{\text{a.s.}} \zeta_\alpha(t_1), \ldots, \zeta_\alpha(t_r), \tag{27}$$

*and for the remaining $m_\star - r$ eigenvectors it holds that*

$$\lambda_{r+1}^{\boldsymbol{T}}, \ldots, \lambda_{m_\star}^{\boldsymbol{T}} \xrightarrow{\text{a.s.}} \zeta_\alpha(\bar{t}_\alpha). \tag{28}$$

As a first result, we derive a bound for $\alpha_k$.

Consider the matrix in equation 24

$$\boldsymbol{G}(y) := \mathbb{E}\left[\frac{(\boldsymbol{z}\boldsymbol{z}^\top - \boldsymbol{I}_m)\mathcal{T}(y)}{t - \mathcal{T}(y)}\right] = \mathbb{E}_y\left[\frac{\mathbb{E}[\boldsymbol{z}\boldsymbol{z}^\top - \boldsymbol{I}_m|y]\mathcal{T}(y)}{t - \mathcal{T}(y)}\right]. \tag{29}$$

It is straightforward to show that it is a diagonal matrix, due to the parity of each function $g_k$. Indeed, given $k \neq h$,

$$G_{kh}(y) = \mathbb{E}[z_k z_h|y] \propto \int e^{-\|\boldsymbol{z}\|^2/2} \exp\left(-\left(y - \sum_{k=1}^m a_k g_k(z_k)\right)^2 / (2\Delta)\right) z_h z_k\, \mathrm{d}\boldsymbol{z} = 0. \tag{30}$$

For simplicity, we denote $G_k(y) := G_{kk}(y)$. Therefore, the solutions of equation 24 coincides with the solutions of

$$\alpha^{-1} = \mathbb{E}_y\left[\frac{\mathcal{T}(y)}{t - \mathcal{T}(y)} G_k(y)\right], \quad k \in [m]. \tag{31}$$

Further, note that, by definition $\bar{t}_\alpha$ is the solution of

$$\alpha^{-1} = \mathbb{E}_y\left[\left(\frac{\mathcal{T}(y)}{t - \mathcal{T}(y)}\right)^2\right]. \tag{32}$$

A spectral transition occurs if, for sample complexity $\alpha$ and index $k \in [m]$, the value $\bar{t}_\alpha$ is also solution of equation 31. Indeed by Theorem 4.2 in Kovačević et al. (2025) – which characterizes the overlap of the principal $r$ eigenvalues, such sample complexity corresponds to the $k$-critical threshold $\alpha_k$. This implies, for $\alpha = \alpha_k^{\text{sp}}$

$$\begin{cases} (\alpha_k^{\text{sp}})^{-1} = \mathbb{E}_y\left[\frac{\mathcal{T}(y)}{t - \mathcal{T}(y)} G_k(y)\right] \\ (\alpha_k^{\text{sp}})^{-1} = \mathbb{E}_y\left[\left(\frac{\mathcal{T}(y)}{t - \mathcal{T}(y)}\right)^2\right]. \end{cases} \tag{33}$$

In order to characterize the threshold, we consider the following expansions for small SNR $a_k\star \ll 1$. Define Z the marginal distribution of $y$ and consider the Fourier representation $(2\pi)^{-1/2}e^{-x^2/2} =$

$(2\pi)^{-1} \int \mathrm{d}\omega e^{i\omega x - \omega^2/2}$, then

$$Z(y) := \frac{1}{\sqrt{2\pi\Delta}} \mathbb{E}_{\boldsymbol{z}} \left[ \exp\left( -\left( y - \sum_{k=1}^{m_\star} a_k^\star g_k(z_k) \right)^2 / (2\Delta) \right) \right] \tag{34}$$

$$= \frac{1}{2\pi} \int \mathrm{d}\omega e^{i\omega y - \omega^2 \Delta/2} \left( \prod_{h \neq k} \mathbb{E}_z \left[ e^{-i\omega a_h^\star g_h(z)} \right] \right) \left[ e^{-i\omega a_k^\star g_k(z)} \right] \tag{35}$$

$$= \frac{1}{2\pi} \int \mathrm{d}\omega e^{i\omega y - \omega^2 \Delta/2} \left( \prod_{h \neq k} \mathbb{E}_z \left[ e^{-i\omega a_h^\star g_h(z)} \right] \right) \left[ 1 + \sum_{\beta \geq 1} \frac{(-i\omega a_k^\star)^\beta}{\beta!} \mathbb{E}_z[g_k^\beta(z)] \right] \tag{36}$$

$$= \frac{1}{2\pi} \int \mathrm{d}\omega e^{i\omega y - \omega^2 \Delta/2} \left( \prod_{h \neq k} \mathbb{E}_z \left[ e^{-i\omega a_h^\star g_h(z)} \right] \right) + O\left( a_k^\star Z'(y) \right), \tag{37}$$

where we have used the identity $\int \mathrm{d}\omega e^{i\omega y} \omega f(\omega) = \frac{\partial}{\partial y} \int \mathrm{d}\omega e^{i\omega y} f(\omega)$. Similarly, one can bound the following quantity

$$\left| G_k(y) + a_k^\star \mathbb{E}_z[g_k(z)(z^2 - 1)] \frac{Z'(y)}{Z(y)} \right| \tag{38}$$

$$= \frac{1}{Z(y)} \left| \mathbb{E}_{\boldsymbol{z} \sim \mathcal{N}(\boldsymbol{0}, \boldsymbol{I}_p)} \left[ P(y|\boldsymbol{z})(z_k^2 - 1) + a_k^\star \mathbb{E}_z[g_k(z)(z^2 - 1)] \partial_y P(y|\boldsymbol{z}) \right] \right| \tag{39}$$

$$= \frac{1}{2\pi Z(y)} \left| \int_{\mathbb{R}} \mathrm{d}\omega e^{i\omega y - \omega^2 \Delta/2} \prod_{h \neq k} \mathbb{E}_z[e^{-i\omega a_h^\star g_h(z)}] \left( \mathbb{E}_z \left[ e^{-i\omega a_k^\star g_k(z)}(z^2 - 1) \right] + i\omega a_k^\star \mathbb{E}_z[g_k(z)(z^2 - 1)] \mathbb{E}_z[e^{-i\omega a_k^\star g_k(z)}] \right) \right| \tag{40}$$

$$= \frac{1}{2\pi Z(y)} \left| \int_{\mathbb{R}} \mathrm{d}\omega e^{i\omega y - \omega^2 \Delta/2} \prod_{h \neq k} \mathbb{E}_z[e^{-i\omega a_h^\star g_h(z)}] O\left( (a_k^\star)^2 \omega^2 \right) \right| \tag{41}$$

$$= \frac{1}{2\pi Z(y)} O\left( (a_k^\star)^2 \left| \frac{\partial^2}{\partial y^2} \int_{\mathbb{R}} \mathrm{d}\omega e^{i\omega y - \omega^2 \Delta/2} \prod_{k \neq h} \mathbb{E}_z[e^{-i\omega a_k^\star g_k(z)}] \right| \right) \tag{42}$$

$$= O\left( (a_k^\star)^2 \left| \frac{Z''(y)}{Z(y)} \right| \right). \tag{43}$$

Note that $\mathbb{E}_z[g_k(z)(z^2 - 1)] \overset{\text{parts}}{=} \mathbb{E}_z[g_k''(z)] \neq 0 \, \forall k \in [m_\star]$ by Assumption 5.1. We now look for $t > \tau$ and $\alpha$ that are solutions of eqs. equation 33. In particular, the oracle $k$-critical threshold in Appendix 5.1 is a lower bound to the spectral one, therefore $\alpha_k^{\text{sp}} = \Omega(\alpha_k^{\text{oracle}})$ and diverges with $k$. The second equation of equation 33 – due to the monotonicity of the RHS, implies that also the solution $t$ diverges with $k$. At leading order in $a_k^\star$ and $t$

$$\begin{cases} (\alpha_k^{\text{sp}})^{-1} \approx -\int \mathrm{d}y Z(y) \frac{\mathcal{T}(y) Z'(y)}{t Z(y)} a_k^\star \mathbb{E}_z[g_k(z)(z^2 - 1)] \approx t^{-1} a_k^\star \mathbb{E}_z[g_k(z)(z^2 - 1)] \mathbb{E}[\mathcal{T}'(y)] \\ (\alpha_k^{\text{sp}})^{-1} \approx t^{-2} \mathbb{E}[\mathcal{T}(y)^2]. \end{cases} \tag{44}$$

The system is thereby solved for

$$t^{-1} = \Theta(a_k^\star), \qquad \alpha_k^{\text{sp}} = \Theta(\alpha_k^{\text{oracle}}). \tag{45}$$

We can now derive the scaling laws for the spectral weighted mean-squared error. Taking $\alpha \gg \alpha_k^{\text{sp}}$, for $k$ large enough, the solution $t_k$ of equation 31 scales as $t_k = \Theta(\alpha a_k^\star)$. As a consequence of Theorem 4.2 in Kovačević et al. (2025), specialized in our setting – where we can exploit the diagonality of $\boldsymbol{G}(y)$, we have that the overlap between the eigenvector $\boldsymbol{v}_k$ corresponding to the eigenvalue $\lambda_k^T$, and the true signal $\boldsymbol{w}_h^\star$, converges, in the high-dimensional limit, to an overlap

$$m_{hk}^2 := \frac{1}{d^2} \langle \boldsymbol{w}_h^\star, \boldsymbol{v}_k \rangle^2 = \delta_{kh} \frac{\zeta_\alpha'(t_k)}{\zeta_\alpha'(t_k) + \frac{\mathrm{d}}{\mathrm{d}t} R_k(t_k)}, \tag{46}$$

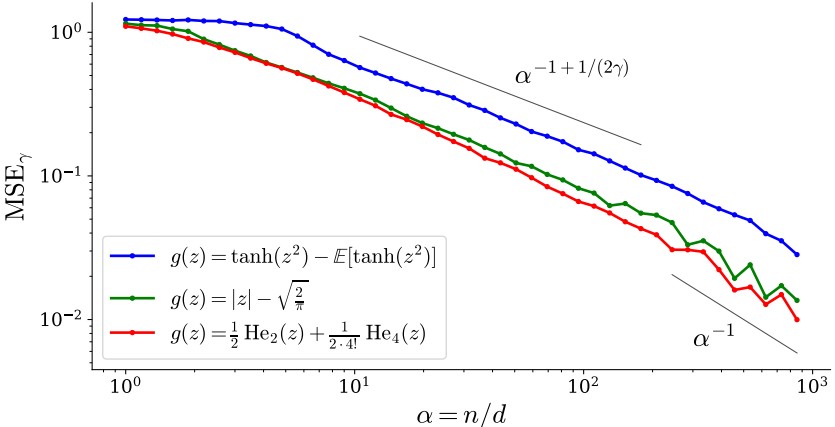

Figure 2: Weighted mean square error $\mathrm{MSE}_\gamma$ – see Def. 2.5 – of the spectral estimator of Def. 3.3 with preprocessing function $\mathcal{T}(y) = y/(1+|y|)$, averaged over 70 instances. The target is given by the hierarchical multi-index model 2.1, with $g_k(z) = g(z)\ \forall k$, stated in the legend, and $a_k^\star \propto k^{-\gamma}$, $\gamma = 1.3$. The covariates dimension is $d = 1000$, the feature space dimension is $m_\star = 10$.

where $R_k(t) = \mathbb{E}[z_k^2 \mathcal{T}(y)/(t - \mathcal{T}(y))]$. For $\alpha \gg \alpha_k^{\mathrm{sp}}$ we have that $t_k \gg 1$ and equation 31 is solved by $t_k = \Theta(a_k^\star \alpha)$. Putting all together

$$m_{hk}^2 = \delta_{kh}\left(1 - \Theta\left(\frac{(\alpha_k^\star)^{-2}}{\alpha}\right)\right). \tag{47}$$

Recall that

$$\mathrm{mse}_k(\boldsymbol{v}_k) = \frac{1}{d^2}\mathbb{E}\left[\|\boldsymbol{v}_k\boldsymbol{v}_k\top\|_{\mathrm{F}}^2 + \|\boldsymbol{w}_k^\star\boldsymbol{w}_k^{\star\top}\|_{\mathrm{F}}^2 - 2\langle\boldsymbol{w}_k^\star, \boldsymbol{v}_k\rangle^2\right] \to 2(1 - m_{kk}^2). \tag{48}$$

By repeating computations analogous to the ones in Appendix 5.1, one finds, setting $\hat{\boldsymbol{W}} = (\boldsymbol{v}_1, \ldots, \boldsymbol{v}_r)^\top$,

$$\mathrm{MMSE}_\gamma(\hat{\boldsymbol{W}} := (\hat{\boldsymbol{W}}^{\mathrm{sp}}, \boldsymbol{0}_{(m_\star - r)\times d})) = \Theta(\mathrm{MMSE}_\gamma^{\mathrm{oracle}}), \tag{49}$$

which proves Theorem 3.2 and 3.5.

### 5.2.1 GD INTERPRETATION OF THE SPECTRAL METHOD

Note that this spectral method also admits an interpretation in terms of a gradient descent algorithm. Indeed, considering a slightly modified loss than the pure square one, this matrix is nothing but the Hessian of the loss Bonnaire et al. (2025). The early dynamics of GD on the first layer of (5), for weights initialized on a sphere with small radius, will thus follow the proposed spectral method. This construction was recently used in Zhang et al. (2025) to show how two-layer neural networks identify features within the first few GD steps in the first layer weights. More precisely, they show that the early dynamics of a suitable gradient-based algorithm, reads as a power iteration

$$\boldsymbol{w}_k^{t+1} = \boldsymbol{w}_k^t - \nabla_{\boldsymbol{w}_k}\frac{1}{n}\sum_{i=1}^n \ell\left(y_i, \boldsymbol{a}^\top\sigma(\boldsymbol{W}^t\boldsymbol{x}_i)\right) \tag{50}$$

$$\approx (\boldsymbol{I}_d - a_k\sigma''(0)\boldsymbol{T})\boldsymbol{w}_k^t, \tag{51}$$

where $\boldsymbol{T}$ has the structure defined in equation 8 with pre-processing $\mathcal{T} = \ell'(\cdot, 0)$, and $\ell'$ denoting the derivative of $\ell$ with respect to its second argument.

## 6 PROOF SKETCH OF THEOREM 3.6

In this section we prove Theorem 3.6. This final result demonstrate that neural networks can efficiently learn the target function, achieving the exact same rates as those of the optimal feature

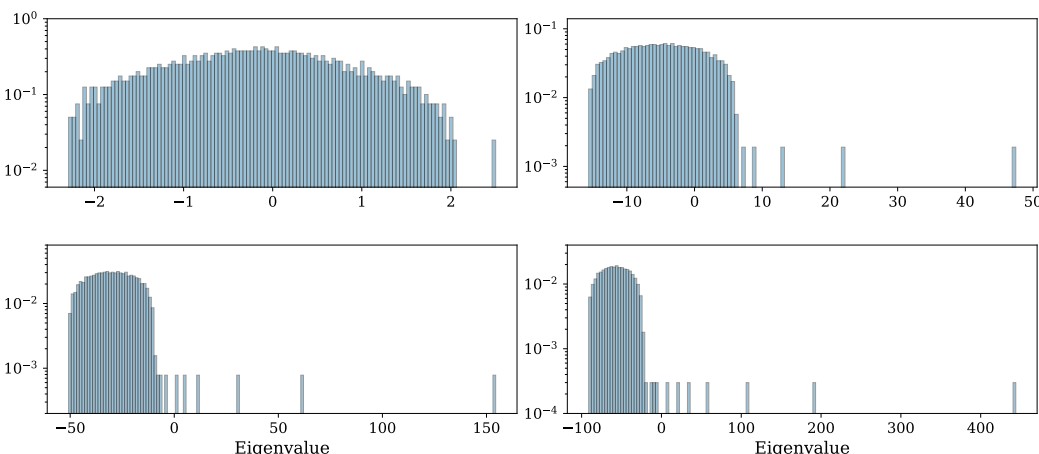

Figure 3: Empirical spectrum density of the matrix $T$ defined in equation 8, with preprocessing function $\mathcal{T}(y) = y/(1+|y|)$, at different sample complexities, highlighting the sequential emergence of concepts as the sample size increases. The target is given by a hierarchical multi-index model 2.1, with $g_k(z) = \frac{1}{2}\mathrm{He}_2(z) + \frac{1}{2\cdot 4!}\mathrm{He}_4(z)$ $\forall k$ and $a_k \propto k^{-\gamma}$, $\gamma = 1.3$. The covariates dimension is $d = 1000$, the feature space dimension is $m_\star = 20$. (**top left**) $\alpha = 5$, (**top right**) $\alpha = 164$, (**bottom left**) $\alpha = 611$, (**bottom right**) $\alpha = 1638$.

subspace recovery. We note that our results do not imply that these rates correspond to the decay of the Bayes risk, defined as $\mathbb{E}\left[(f_\star(\boldsymbol{x}) - \mathbb{E}[f_\star(\boldsymbol{x})|\boldsymbol{x}, \mathcal{D}])^2\right]$, and which constitutes a lower bound for the risk $R(\boldsymbol{\Theta})$ in equation 6. Nevertheless, we notice that under the hypothesis of a *fully specialized* Bayes-optimal estimator—i.e., assuming $d^{-1}\hat{\mathbf{W}}^{\mathrm{Bayes}}(\mathbf{W}^\star)^\top$ converges to a diagonal matrix in the high-dimensional limit—the risk for a network with matching architecture and Bayes weights, $f(\mathbf{x}) = \sum_{k=1}^{m_\star} a_k^\star g_k(\hat{\boldsymbol{w}}_k^{\mathrm{Bayes}})$, achieves the rates established in Theorem 3.2. We leave a more precise analysis of the optimal risk as a future direction.

---

**Algorithm 1** Spectral Initialization and Ridge Training
___

**Input:** Dataset $\mathcal{D} = (\boldsymbol{X}, \boldsymbol{y}) \in \mathbb{R}^{2n \times d} \times \mathbb{R}^{2n}$, hidden layer width $m$, regularization $\lambda$.
**Output:** Network parameters $\boldsymbol{W} \in \mathbb{R}^{p \times d}$, $\boldsymbol{a} \in \mathbb{R}^p$.

*1. Data Splitting*
Partition the dataset $\mathcal{D}$ into two disjoint sets $\mathcal{D}_1$ and $\mathcal{D}_2$ such that $|\mathcal{D}_1| = |\mathcal{D}_2| = n$.

*2. Feature Learning (Spectral Initialization on $\mathcal{D}_1$)*
Compute the spectral estimator $\hat{\boldsymbol{W}}^{\mathrm{sp}} \in \mathbb{R}^{r \times d}$ Def. 3.3 using $\mathcal{D}_1$.
Sample a random matrix $\boldsymbol{Z} \in \mathbb{R}^{p \times r}$ with i.i.d. entries drawn from $\mathcal{N}(0, r^{-1})$.
Set first-layer weights: $\boldsymbol{W} = \boldsymbol{Z}\hat{\boldsymbol{W}}$.

*3. Readout Training (Ridge Regression on $\mathcal{D}_2$)*
Compute the feature matrix $\Psi \in \mathbb{R}^{n \times p}$ on $\mathcal{D}_2$, where $\Psi_{ij} = \sigma(\langle \boldsymbol{w}_j, \boldsymbol{x}_i^{(2)} \rangle + b_j)$, $b_j \sim \mathcal{N}(0, 1)$.
Solve for the second-layer weights:
$\quad \boldsymbol{a} = (\Psi^\top \Psi + n\lambda \boldsymbol{I}_p)^{-1} \Psi^\top \boldsymbol{y}^{(2)}$

**return** $\boldsymbol{W}, \boldsymbol{a}$

---

Denote $\hat{\boldsymbol{W}}^{\mathrm{sp}} \in \mathbb{R}^{r \times d}$ the spectral estimator defined in 3.3. In Appendix 5.2 we have shown that each column $k$ correlates with the signal component $\boldsymbol{w}_k^\star$ only, with squared overlap $m_k = 1 - \Theta((\alpha_k^\star)^{-2}/\alpha)$. Denote $\boldsymbol{v}_k$ as the $k^{\mathrm{th}}$ column of the spectral estimator and $\eta_k^2 := (\alpha_k^\star)^2/\alpha \ll 1$. Up

to negligible corrections $O(\eta_k^2)$

$$\boldsymbol{w}_k^\star = \boldsymbol{v}_k + \eta_k \boldsymbol{\xi}_k, \tag{52}$$

with $\boldsymbol{\xi}_k$ a unit vector orthogonal to all $\{\boldsymbol{v}_h\}_{h \in [r]}$. Given a covariate $\boldsymbol{x} \in \mathbb{R}^d$, define the projected inputs $\boldsymbol{s} \in \mathbb{R}^r$ such that $s_k = \langle \boldsymbol{v}_k, \boldsymbol{x} \rangle$, $k \in [r]$. The first layer pre-activations are given by

$$\boldsymbol{W}\boldsymbol{x} = \boldsymbol{Z}\hat{\boldsymbol{W}}^{\text{sp}}\boldsymbol{x} = \boldsymbol{Z}\boldsymbol{s}. \tag{53}$$

Similarly, up to negligible corrections $O(\eta_k^2)$, the target function

$$f_\star(\boldsymbol{x}) = \sum_{k=1}^r a_k^\star g_k(s_k + \eta_k \langle \boldsymbol{\xi}_k, \boldsymbol{x} \rangle) + \sum_{k=r+1}^{m_\star} a_k^\star g_k(\langle \boldsymbol{w}_k^\star, \boldsymbol{x} \rangle) \tag{54}$$

$$= \sum_{k=1}^r a_k^\star g_k(s_k) + \sum_{k=1}^r a_k^\star g_k'(s_k)\eta_k \langle \boldsymbol{\xi}_k, \boldsymbol{x} \rangle + \sum_{k=r+1}^{m_\star} a_k^\star g_k(\langle \boldsymbol{w}_k^\star, \boldsymbol{x} \rangle). \tag{55}$$

Due to the orthogonality between $\boldsymbol{\xi}_k$ and $\boldsymbol{v}_h$, for any $h, k \in [r]$, the variables $\langle \boldsymbol{x}, \boldsymbol{\xi}_k \rangle$ and $s_h$ are independent centered Gaussian variables. With respect to the projected input space, the effective target function is

$$f_\star^{\text{eff}}(\boldsymbol{s}) := \mathbb{E}[f_\star(\boldsymbol{x})|\boldsymbol{s}] = \sum_{i=1}^r a_k^\star g_k(s_k). \tag{56}$$

Then, the readout training in Algorithm 1 corresponds to random feature ridge regression Rahimi & Recht (2007) with weights $\boldsymbol{Z} = (\boldsymbol{z}_1, \ldots, \boldsymbol{z}_p)^\top$ on the projected covariates $\boldsymbol{s}_i = \hat{\boldsymbol{W}}^{\text{sp}}\boldsymbol{x}_i^{(2)}$, $i \in [n]$, where $\{\boldsymbol{x}_i^{(2)}\}_{i \in [n]}$ are the covariates in the dataset $\mathcal{D}_2$:[3]

$$\hat{\boldsymbol{a}} = \arg\min_{\boldsymbol{a} \in \mathbb{R}^p} \frac{1}{2n} \sum_{i=1}^n \left( y_i - \sum_{j=1}^p a_j \sigma(\langle \boldsymbol{z}_j, \boldsymbol{s}_i \rangle) \right)^2 + \frac{\lambda}{2} \|\boldsymbol{a}\|^2 \tag{57}$$

Since $\sigma$ is bounded and continuous, we can apply Theorem 1 in Rudi & Rosasco (2017)[4], choosing $p = \omega(n^{1/2})$, $\lambda = \Theta(n^{-1/2})$, so that we obtain that the risk equation 6 $R(\hat{\boldsymbol{a}}, \boldsymbol{W})$ satisfies

$$R(\hat{\boldsymbol{a}}, \boldsymbol{W}) - R_{f_{\mathcal{H}}} = O(n^{-1/2}), \tag{58}$$

where, given the *reproducing kernel Hibert space* (RKHS) $\mathcal{H}$ associated to the kernel $K(\boldsymbol{s}, \boldsymbol{s}') = \mathbb{E}_{\boldsymbol{z}, b}[\sigma(\boldsymbol{s}^\top \boldsymbol{s} + b)\sigma(\boldsymbol{w}^\top \boldsymbol{s}' + b)]$,

$$R_{f_{\mathcal{H}}} = \min_{f \in \mathcal{H}} \mathbb{E}_{y, \boldsymbol{s}}[(f(\boldsymbol{s}) - y)^2]. \tag{59}$$

Such irreducible risk is lower-bounded by

$$R_\star = \mathbb{E}[(f_\star^{\text{eff}}(\boldsymbol{s}) - y)^2] = \Theta\left( \sum_{k=1}^r (a_k^\star)^2 \eta_k^2 + \sum_{k=r+1}^{m_\star} (a_k^\star)^2 \right) = \Theta(\text{MMSE}_\gamma), \tag{60}$$

where, as for the spectral method mean-squared error, the two terms correspond to the approximation error of learned features and underfitting of unlearned ones. If $f_\star^{\text{eff}} \in \mathcal{H}$, the argument is complete. By assumption, the function $\sigma$ allows for a decomposition in Hermite polynomials

$$\sigma(z) = \sum_{\beta \geq 0} \frac{\sigma_\beta}{\beta!} \text{He}_\beta(z), \quad \sigma_\beta := \frac{1}{\beta!} \mathbb{E}_z[\text{He}_\beta(z)\sigma(z)]. \tag{61}$$

---

[3] In order to simplify the notation, we denote the labels in $\mathcal{D}_2$ as $y_i := y_i^{(2)}$.

[4] Note that the assumption of bounded outputs in Theorem 1 is relaxed in the Appendix of Rudi & Rosasco (2017), see Assumption 4. Further, the analysis in Rudi & Rosasco (2017) allows for a more refined error rate in our setting. However, the current result is already subleading with respect to the rates in Theorem 3.2.

Further, being bounded, it cannot be a polynomial of finite degree, as there is no $\beta_0$ such that $\sigma_\beta = 0$ for all $\beta \geq \beta_0$, excluding the trivial case of constant functions. Considering the Taylor expansion of Hermite polynomials

$$\mathrm{He}_\beta(z + b) = \sum_{\zeta=0}^{\beta} \binom{\beta}{\zeta} b^{\beta-\zeta} \mathrm{He}_\zeta(z), \tag{62}$$

with $\binom{\beta}{\zeta} = \frac{\beta!}{\zeta!(\beta-\zeta)!}$, we find that the kernel $K$ is given by

$$K(\boldsymbol{s}, \boldsymbol{s}') = \mathbb{E}_{\boldsymbol{z},b}[\sigma(\boldsymbol{z}^\top \boldsymbol{s} + b)\sigma(\boldsymbol{z}^\top \boldsymbol{s}' + b)] \tag{63}$$

$$= \mathbb{E}_{\boldsymbol{z},b} \sum_{\beta,\beta' \geq 0} \sum_{\zeta=0}^{\beta} \sum_{\zeta'=0}^{\beta'} \frac{\sigma_\beta \sigma_{\beta'}}{\zeta!\zeta'!(\beta-\zeta)!(\beta'-\zeta')!} b^{\beta-\zeta} b^{\beta'-\zeta'} \mathrm{He}_\zeta(\langle \boldsymbol{z}, \boldsymbol{s} \rangle) \mathrm{He}_{\zeta'}(\langle \boldsymbol{z}, \boldsymbol{s}' \rangle) \tag{64}$$

By Proposition 11.31 in O'Donnell (2014),

$$K(\boldsymbol{s}, \boldsymbol{s}') = \mathbb{E}_b \sum_{\beta,\beta' \geq 0} \sum_{\zeta=0}^{\min(\beta,\beta')} \frac{\sigma_\beta \sigma_{\beta'}}{\zeta!(\beta-\zeta)!(\beta'-\zeta)!} b^{\beta+\beta'-2\zeta} \frac{\langle \boldsymbol{s}, \boldsymbol{s}' \rangle^\zeta}{r^\zeta} \tag{65}$$

$$= \mathbb{E}_b \sum_{\zeta=0}^{\infty} \sum_{\beta,\beta' \geq \zeta} \frac{\sigma_\beta \sigma_{\beta'}}{\zeta!(\beta-\zeta)!(\beta'-\zeta)!} b^{\beta+\beta'-2\zeta} \frac{\langle \boldsymbol{s}, \boldsymbol{s}' \rangle^\zeta}{r^\zeta} \tag{66}$$

$$= . \sum_{\zeta=0}^{\infty} \mathbb{E}_b \underbrace{\left( \sum_{\beta \geq \zeta} \frac{\sigma_\beta}{\zeta!(\beta-\zeta)!} b^{\beta-\zeta} \right)^2}_{} \frac{\langle \boldsymbol{s}, \boldsymbol{s}' \rangle^\zeta}{r^\zeta} \tag{67}$$

The bracketed term is always strictly positive as $\sum_{\beta \geq \zeta} \frac{\sigma_\beta}{\zeta!(\beta-\zeta)!} b^{\beta-\zeta}$ in $b$ cannot be identically zero. This would require $\sigma_\beta = 0$ for all $\beta > \zeta$, which contradicts the hypothesis of boundedness. Expanding the term $\langle \boldsymbol{s}, \boldsymbol{s}' \rangle$, it follows that the kernel admits a decomposition as $K(\boldsymbol{s}, \boldsymbol{s}') = \Phi(\boldsymbol{s})^\top \Phi(\boldsymbol{s}')$, where $\Phi$ is a feature map with components given by multivariate monomials $\lambda_{\boldsymbol{\beta}} s_1^{\beta_1} \dots s_r^{\beta_r}$, for some $\lambda_{\boldsymbol{\beta}} > 0$ for all degrees $|\boldsymbol{\beta}|$. Theorem 4.21 in Christmann & Steinwart (2008) readily implies that the RKHS of $K$ includes all finite degree polynomials, completing our argument.

## 7 USEFUL RESULTS ON SINGLE-INDEX MODELS

In this section we present various results applied in the derivation of the lower bound to $\mathrm{MMSE}_\gamma$ in Appendix 5.1. In particular, we focus on the following single-index model setting.

**Definition 7.1** *Let* $\boldsymbol{w}^\star \sim \mathcal{N}(\boldsymbol{0}_d, \boldsymbol{I}_d)$, $g : \mathbb{R} \to \mathbb{R}$ *satisfy Assumption 5.1. Consider the supervised learning problem of estimating* $\boldsymbol{w}_\star$ *from* $n$ *i.i.d. observations* $\mathcal{D} = \{(\boldsymbol{x}_i, y_i) \in \mathbb{R}^{d \times 1} : i \in [n]\}$ *generated as*

$$\boldsymbol{x}_i \sim \mathcal{N}(\boldsymbol{0}, \boldsymbol{I}_d), \qquad y_i = \sqrt{\lambda} g(\langle \boldsymbol{x}_i, \boldsymbol{w}^\star \rangle) + \xi_i, \tag{68}$$

*where* $\lambda \geq 0$ *is the signal-to-noise ratio (SNR) and* $\xi_i \sim \mathcal{N}(0, 1)$ *is additive noise.*

Given an estimator $\hat{\boldsymbol{w}}$ of $\boldsymbol{w}^\star$ that is a function of the dataset, we evaluate its estimation error using the following matrix-MMSE:

$$\mathrm{mse}(\boldsymbol{w}) := \frac{1}{d^2} \mathbb{E}[\|\boldsymbol{w}\boldsymbol{w}^\top - \boldsymbol{w}^\star \boldsymbol{w}^{\star\top}\|_F^2], \tag{69}$$

where $\mathbb{E}$ computes the expected value with respect to the joint distributions of $\mathcal{D}$ and $\boldsymbol{w}_\star$. A lower bound to this quantity is given by the following optimal matrix-MSE

$$\mathrm{mmse} := \frac{1}{d^2} \mathbb{E} \left[ \|\mathbb{E}[\boldsymbol{w}\boldsymbol{w}|\mathcal{D}] - \boldsymbol{w}^\star \boldsymbol{w}^{\star\top}\|_F^2 \right] = \underset{\boldsymbol{Q} \in \mathbb{R}^{d \times d}}{\arg\min} \mathbb{E} \left[ \|\boldsymbol{Q} - \boldsymbol{w}^\star \boldsymbol{w}^{\star\top}\|_F^2 \right]. \tag{70}$$

Intuitively, such optimal estimation error decreases with the SNR.

**Lemma 7.2** *In the setting defined in 7.1, the optimal matrix-MSE satisfies*

$$\frac{\partial}{\partial\lambda}\text{mmse} < 0. \tag{71}$$

The proof follows from direct computation. We provide here a condensed derivation.

$$\text{mmse} = \mathbb{E}\left[\left\|\boldsymbol{w}\boldsymbol{w}^T\right\|_F^2\right] - \mathbb{E}_{\boldsymbol{y}}\left[\left\|\mathbb{E}[\boldsymbol{w}\boldsymbol{w}^T \mid \boldsymbol{y}]\right\|_F^2\right]. \tag{72}$$

Since the first term is independent of $\lambda$, it suffices to show that the second term is non-decreasing. Let us denote the posterior mean as $\langle\boldsymbol{w}\boldsymbol{w}^T\rangle_\lambda \equiv \mathbb{E}[\boldsymbol{w}\boldsymbol{w}^T \mid \boldsymbol{y}]$. We compute its derivative with respect to $\lambda$:

$$\frac{d}{d\lambda}\mathbb{E}_{\boldsymbol{y}}\left[\|\langle\boldsymbol{w}\boldsymbol{w}^T\rangle_\lambda\|_F^2\right] = \int d\boldsymbol{y}\,\frac{\partial}{\partial\lambda}\left(\mathsf{P}(\boldsymbol{y})\|\langle\boldsymbol{w}\boldsymbol{w}^T\rangle_\lambda\|_F^2\right). \tag{73}$$

Using the explicit form of $\mathsf{P}(\boldsymbol{y}|\boldsymbol{w}) \propto \exp(-\frac{1}{2}\|\boldsymbol{y} - \sqrt{\lambda}g(\boldsymbol{X}\boldsymbol{w})\|^2)$, the derivative $\partial_\lambda \ln \mathsf{P}(\boldsymbol{y})$ introduces terms involving $\boldsymbol{y}^T\langle g(\boldsymbol{X}\boldsymbol{w})\rangle_\lambda$. We handle these terms via the multivariate Stein's Lemma (corresponding to integration by parts), which states that for the Gaussian measure, $\mathbb{E}_{\boldsymbol{y}}[\boldsymbol{y}^T\mathbf{f}(\boldsymbol{y})] = \mathbb{E}_{\boldsymbol{y}}[\text{Tr}(\nabla\mathbf{f}) + \sqrt{\lambda}\langle h(\boldsymbol{X}\boldsymbol{w})\rangle_\lambda^T\mathbf{f}]$.

Applying this identity results in cancellations of the lower-order terms. The surviving term is proportional to the gradient of the estimator with respect to the observations $\nabla_{\boldsymbol{y}}\langle\boldsymbol{w}\boldsymbol{w}^T\rangle_\lambda = \sqrt{\lambda}\,\text{Cov}(\boldsymbol{w}\boldsymbol{w}^T, g(\boldsymbol{X}\boldsymbol{w}) \mid \boldsymbol{y})$. Therefore,

$$\frac{d}{d\lambda}\mathbb{E}_{\boldsymbol{y}}\left[\|\langle\boldsymbol{w}\boldsymbol{w}^T\rangle_\lambda\|_F^2\right] = \mathbb{E}_{\boldsymbol{y}}\left[\left\|\text{Cov}\left(\boldsymbol{w}\boldsymbol{w}^T, g(\boldsymbol{X}\boldsymbol{w}) \mid \boldsymbol{y}\right)\right\|_F^2\right] \implies \frac{\partial}{\partial\lambda}\text{mmse} < 0. \tag{74}$$

We now consider the high-dimensional limit $n, d \to \infty$ with fixed ratio $\alpha = n/d$, referred to as the sample complexity. The following theorem specializes Theorems 1 and 2 of Barbier et al. (2019) to the setting of interest.

**Theorem 7.3 (Barbier et al. (2019))** *Consider the setting defined in 7.1. Then, in the limit $n, d \to \infty$, with fixed ratio $n/d = \alpha$,*

$$\frac{1}{d^2}\mathbb{E}\|\boldsymbol{w}^\star\boldsymbol{w}^{\star\top} - \mathbb{E}[\boldsymbol{w}\boldsymbol{w}\top \mid \mathcal{D}]\|_F^2 \to 1 - m^2, \tag{75}$$

*and, given $\boldsymbol{w} \sim \mathsf{P}(\cdot|\mathcal{D})$,*

$$\frac{1}{d}|\boldsymbol{w}^\top\boldsymbol{w}^\star| \xrightarrow{\mathbb{P}} m, \tag{76}$$

*where $m = m(\alpha)$ is the maximizer of*

$$\sup_{m\in[0,1]} f_{\text{RS}}(m), \quad f_{\text{RS}}(m) := \{m + \log(1 - m) + 2\alpha\Psi_{\text{out}}(m)\}, \tag{77}$$

$$\Psi_{\text{out}}(m) := \mathbb{E}_{W,V,Y}\log\mathbb{E}_{w\sim\mathcal{N}(0,1)}\left[\mathsf{P}(Y|\sqrt{m}V + \sqrt{1-m}w)\right], \tag{78}$$

*with $V, W \sim \mathcal{N}(0, 1)$, $Y \sim \mathsf{P}(\cdot|\sqrt{m}V + \sqrt{1-m}W)$.*

As a direct consequence of the theorem, the information-theoretic weak recovery threshold $\alpha^{\text{IT}}$ is the smallest sample complexity $\alpha$ such that the maximizer of the free entropy equation 77 is $m \neq 0$. Equivalently, $\alpha^{\text{IT}}$ is the smallest $\alpha$ such that $\text{mmse} < 1$. Lemma 7.2 readily implies that $\alpha^{\text{IT}} = \alpha^{\text{IT}}(\lambda)$ is decreasing with the SNR.

Finally, we characterize the IT weak recovery threshold in the limit of small SNR. By expanding equation 78 around $\lambda = 0$, we derive the following Corollary.

**Corollary 7.4 (IT weak recovery threshold in the large noise regime)** *Consider the setting of Theorem 7.3. Then, in the limit $\lambda \to 0$, the information theoretic weak-recovery threshold satisfies*

$$\alpha^{\text{IT}} = \Theta(\lambda^{-1}) \tag{79}$$

*and, for $\alpha \to \infty$, and fixed $\lambda$ the optimal matrix-MSE scales as*

$$\text{mmse} = O\left(\frac{1}{\lambda\alpha}\right). \tag{80}$$

In order to simplify the notation, we introduce the shorthand $\phi_\beta(V; m) = \mathbb{E}_{\boldsymbol{w}}[g^\beta(\sqrt{m}V + \sqrt{1-m}w)]$.

$$\mathbb{E}_w\left[\mathsf{P}(Y = y|\sqrt{m}V + \sqrt{1-m}w)\right] = \frac{1}{\sqrt{2\pi}}\mathbb{E}_w\left[\exp\left(-\frac{(y - \sqrt{\lambda}(\sqrt{m}V + \sqrt{1-m}w))^2}{2}\right)\right] \tag{81}$$

$$= \frac{e^{-y^2/2}}{\sqrt{2\pi}}\left(1 + \sum_{\beta \geq 1}\frac{\lambda^{\beta/2}}{\beta!}\phi_\beta(V; m)\mathrm{He}_\beta(y)\right) \tag{82}$$

and therefore, expanding $\mathsf{P}(y|\sqrt{m}V + \sqrt{1-m}W)$ in a similar fashion and leveraging the expansion $\log(1 + x) = -\sum_{k \geq 0}(-1)^k x^k/k$, up to constant terms with respect to $m$,

$$\Psi_{\mathrm{out}}(m) = \mathbb{E}_{V,W,Y}\left[-\frac{y^2}{2} + \sum_{\beta \geq 1}\frac{\lambda^{\beta/2}}{\beta!}\phi_\beta(V; m)\mathrm{He}_\beta(y) - \frac{1}{2}\left(\sum_{\beta \geq 1}\frac{\lambda^{\beta/2}}{\beta!}\phi_\beta(V; m)\mathrm{He}_\beta(y)\right)^2\right] + O(\lambda^{3/2}) \tag{83}$$

$$= \mathbb{E}_V\int\frac{e^{-y^2/2}}{\sqrt{2\pi}}\left(-\frac{y^2}{2} + \sum_{\beta \geq 1}\frac{\lambda^{\beta/2}}{\beta!}\phi_\beta(V; m)\mathrm{He}_\beta(y) - \frac{\lambda}{2}\phi_1^2(V; m)y^2\right)\mathrm{d}y + \tag{84}$$

$$+ \mathbb{E}_V\int\frac{e^{-y^2/2}}{\sqrt{2\pi}}\left(\left(-\frac{y^2}{2} + \sum_{\beta \geq 1}\frac{\lambda^{\beta/2}}{\beta!}\phi_\beta(V; m)\mathrm{He}_\beta(y)\right)\sum_{\beta \geq 1}\frac{\lambda^{\beta/2}}{\beta!}\phi_\beta(V; m)\mathrm{He}_\beta(y)\right)\mathrm{d}y + O(\lambda^{3/2})$$

$$= -\frac{1}{2} + \frac{\lambda}{2}\mathbb{E}_V[\phi_1^2(V; m)] - \frac{\lambda}{2}\mathbb{E}_V[\phi_2(V; m)] + O(\lambda^{3/2}) \tag{85}$$

$$= \frac{1}{2}\left(-1 - \lambda\mathbb{E}_{z \sim \mathcal{N}(0,1)}[g^2(z)] + \lambda\mathbb{E}_{(z_1,z_2) \sim \mathcal{N}(\mathbf{0}_2, \boldsymbol{C})}[g(z_1)g(z_2)]\right) + O(\lambda^{3/2}) \tag{86}$$

with

$$\boldsymbol{C} = \begin{pmatrix} 1 & m \\ m & 1 \end{pmatrix}. \tag{87}$$

In the above we used

$$\mathbb{E}_V[\phi_2(V; m)] = \mathbb{E}_{W,V}[g^2(\sqrt{m}V + \sqrt{1-m}W)] = \mathbb{E}_z[g^2(z)], \tag{88}$$

$$\mathbb{E}_V[\phi_1^2(V; m)] = \mathbb{E}_V[\mathbb{E}_W[g(\sqrt{m}V + \sqrt{1-m}W)]\mathbb{E}_{W'}[g(\sqrt{m}V + \sqrt{1-m}W')]] \tag{89}$$

$$= \mathbb{E}_{(z_1,z_2) \sim \mathcal{N}(\mathbf{0}_2, \boldsymbol{C})}[g(z_1)g(z_2)]. \tag{90}$$

Assumption 5.1 ensures that $g$ can be decomposed in the Hermite basis as

$$g(z) = \sum_{k \geq 0}\frac{c_k}{k!}\mathrm{He}_k(z), \qquad c_k := \frac{1}{k!}\mathbb{E}_{z \sim \mathcal{N}(0,1)}\left[g(z)\mathrm{He}_k(z)\right]. \tag{91}$$

Leveraging Proposition 11.31 in O'Donnell (2014),

$$\mathbb{E}_{(z_1,z_2) \sim \mathcal{N}(\mathbf{0}_2, \boldsymbol{C})}[g(z_1)g(z_2)] = \sum_{k \geq 0}\frac{c_k^2}{k!}m^k \in [0, \mathbb{E}_z[g^2(z)]], \tag{92}$$

which is a non-decreasing function of $m \in [0, 1]$. Note that, since $g$ has generative exponent 2, and $\mathbb{E}_z[g(z)] = 0$, we have that $c_0 = c_1 = 0$ necessarily. Moreover, by Assumption 5.1, $c_2 \neq 0$ and bounded. Therefore, we are interested in maximizing the quantity[5]

$$f_{\mathrm{RS}}(m) := m + \log(1 - m) + \alpha\lambda\sum_{k \geq 2}\frac{c_k^2}{k!}m^k + O(\alpha\lambda^{3/2}) \tag{93}$$

$$= \frac{1}{2}\left(\alpha\lambda c_2^2 - 1\right)m^2 + \sum_{k > 2}\left(\frac{\alpha\lambda}{k!}c_k^2 - \frac{1}{k}\right)m^k + O(\alpha\lambda^{3/2}), \tag{94}$$

---

[5]Note that we are neglecting constant terms with respect to $m$.

where we have expanded $\log(1-m)$. For $\alpha > \lambda^{-1}c_2^{-2}$, $m = 0$ is a minimum of the free entropy, therefore $\alpha^{\mathrm{IT}} \leq \lambda^{-1}c_2^{-2}$. Denote

$$D := \inf_{m \in (0,1]} \frac{-m - \log(1-m)}{\sum_k \frac{c_k^2}{k!} m^k}, \tag{95}$$

which is strictly positive and well-defined. Note that

$$\lim_{m \to 0^+} \frac{-m - \log(1-m)}{\sum_k \frac{c_k^2}{k!} m^k} = c_2^{-2} \implies \frac{1}{2\mathbb{E}_z[g^2(z)]} \leq D \leq c_2^{-2} = \frac{1}{2\mathbb{E}_z[g''(z)]}. \tag{96}$$

Then, for all $\alpha < D\lambda^{-1}$, $f(m \neq 0) < f(0) = 0$, *i.e.* $m = 0$ is the global maximizer and

$$D \leq \alpha^{\mathrm{IT}}\lambda \leq c_2^{-2} \implies \alpha^{\mathrm{IT}} = \Theta(\lambda^{-1}), \tag{97}$$

For the second result, we first consider the setting $\lambda \to 0$ and $\alpha \to \infty$, with $\alpha \gg \lambda$. There exists a non-zero maximizer $m$, which satisfies[6]

$$\frac{\mathrm{d}}{\mathrm{d}m} f_{\mathrm{RS}}(m) \stackrel{!}{=} 0 \implies \frac{1}{1-m} = \alpha\lambda \sum_{k \geq 2} \frac{c_k^2}{k!} m^{k-2} \implies \sum_{k \geq 2} \frac{c_k^2}{k!}\left(m^{k-2} - m^{k-1}\right) = \frac{1}{\alpha\lambda}, \tag{98}$$

The equation is solved by $m = 1 - \Theta\left(\frac{1}{\alpha\lambda}\right)$. Theorem 7.3 implies that, in this regime, $\mathrm{mmse}(\lambda) = \Theta((\alpha\lambda)^{-1})$. Together Lemma 7.2, the result for arbitrary $\lambda$ is proved.

---

[6]In the following we retain the leading terms in the free entropy expansion.

