# OpenReview forum: "Optimal scaling laws in learning hierarchical multi-index models"
_ICLR.cc/2026/Workshop/Sci4DL — Sci4DL 2026_

### Official Review · Reviewer_8eSd · 2026-02-27

**Fit:** 3
**Significance:** 3
**Confidence:** 2

**Summary:**

Overall I really like this work and recommend it as a full talk.

The authors ask theoretical scaffolding to understand feature learning in a two-layer neural network trained on hierarchical multi-index regression targets. The authors derive the optimal prediction errors for a model trained on n samples with a defined hierarchy and obtain a scaling law for recovering the latent subspace span as a function of sample complexity $\alpha$, $\gamma$ and $m_*$. Furthermore, the authors show that an explicit spectral method can achieve the optimal prediction error derived above, agnostic of the underlying data distribution, and thus the classification readout can be learned without a statistical bottleneck.

**Strengths:**

This theory seeks to understand critical experimental observations in feature learning process described by prior studies and connects them to the spectral structures observed in learned network weights. The theroetical development is strong with quantitative predictions that describe when and how directions emerge in learned representations, as well as predicting sharp phase transitions as the sample size increases. The hierarchical multi index data model is particularly elegant and useful as it allows control over sparsity and complexity of the representations.

**Suggestions:**

It would be great to see more direct tests of the predicted phase transitions and rates across a wider set of settings (such as different link functions and noise levels) and a discussion on what could (strongly) falsify the theory in practice.

---

### Official Review · Reviewer_A3Pn · 2026-02-28

**Fit:** 3
**Significance:** 2
**Confidence:** 2

**Summary:**

This paper studies a learning problem called "hierarchical multi-index models" where the ground truth regression function is a weighted sum (with weights that decay according to a power law) of $m_*$ of (possibly nonlinearly) transformed projections of the $d$-dimensional input signal.
The author's derive Bayes-optimal scaling laws for the estimation of the target function in the  regime where $\frac{n}{d}, m_* >> 1$, define a spectral estimator which can achieve this bound, and design a specific training procedure by which two layer neural networks can recover this solution.
The theory predicts phase transitions  (plateaus and sharp increases) in performance as a function of the sample complexity; the authors draw a connection between this model and empirical observations of "concept emergence" during the training of large neural networks such as language models.

**Strengths:**

- Deriving optimal scaling laws for a well established model of feature learning (even if only a generalization of previous work) is a valuable contribution, and has the potential to mechanistically explain some important empirical observations.
- Going beyond a theoretical optimum and designing an algorithm that can achieve these bounds strengthens the potential for mechanistic explanation referenced above.

**Suggestions:**

- The connection to Schaeffer et al. needs some clarification. My recollection of this work is that it argues that many examples of sudden concept emergence are artifacts of evaluation design choices rather than a genuine reflection of model capability. If this is correct it does not seem to be directly relevant to the setting considered here.
- The neural network training procedure that recovers optimal scaling law performance is somewhat non-standard, and while the author's argue it can be connected to low learning rate gradient descent it would be nice to have direct comparisons between the proposed optimal algorithm and more standard training procedures.

---

### Meta-Review · Area_Chair_v3c2 · 2026-02-28

**Recommendation:** Accept

**Metareview:**

Reviewers liked this paper, and I recommend acceptance.

---

### Decision · Program_Chairs · 2026-03-02

Accept